# Brief Communication: Weak correlation between building damage and loss of life from landslides

Maximillian Van Wyk de Vries[1,2,3], Alexandre Dunant[4,5], Amy L. Johnson[6], Erin L. Harvey[4], Sihan Li[7], Katherine Arrell[8], Jeevan Baniya[9], Dipak Basnet[9], Gopi K. Basyal[10], Nyima Dorjee Bhotia[9], Simon J. Dadson[3], Alexander L. Densmore[4], Tek Bahadur Dong[9], Mark E. Kincey[11], Katie Oven[8], Anuradha Puri[9], and Nick J. Rosser[4]

[1]Department of Geography, University of Cambridge, Cambridge CB2 3EL, UK.
[2]Department of Earth Sciences, University of Cambridge, Cambridge CB3 0EZ, UK.
[3]School of Geography and the Environment, University of Oxford, Oxford OX1 3QY, UK.
[4]Department of Geography, Durham University, Lower Mountjoy, South Rd, Durham DH13LE, UK.
[5]Center for Climate Change and Transformation, Eurac Research, Bolzano 39100, Italy.
[6]Department of Government and Sociology, Georgia College & State University, Milledgeville 31061, US.
[7]Department of Geography, University of Sheffield, Winter St, Sheffield S37ND, UK.
[8]Department of Geography and Environmental Sciences, Northumbria University, City Campus, Newcastle upon Tyne, NE1 7RU, UK.
[9]Social Science Baha, 345 Ramchandra Marg, Battisputali, Kathmandu, Nepal.
[10]National Society for Earthquake Technology, Sainbu Bhainsepati Residential Area, Lalitpur 13775, Nepal.
[11]School of Geography, Politics and Sociology, Newcastle University, Newcastle upon Tyne NE17RX, UK.

**Correspondence:** M. Van Wyk de Vries (msv27@cam.ac.uk)

**Abstract.** Mapping exposure to landslides is necessary to mitigate risk and increase resilience. Exposure maps can be constructed from building databases, akin to seismic risk assessments, but there has been little investigation of the predictive relationship between building damage from landslides and risk to human life. Our study investigates this relationship globally and in Nepal (47,213 and 5,664 landslides, respectively). While a correlation exists for nationwide totals ($R^2$=0.75), it is negligible for individual events ($R^2$=0.025). It is important to not construct landslide exposure maps from building datasets alone, else building damage may be inadvertently prioritised over human lives in disaster planning.

## 1 Introduction

### 1.1 Landslides and landslide risk

Landslides cause thousands of deaths each year across a wide range of geographic environments (Petley, 2012; Kennedy et al., 2015), and are preconditioned and triggered by a wide range of anthropogenic (e.g. road cutting) and physical (e.g. earthquakes and intense rainfall) processes (e.g. van Westen et al., 2006). Landslide risk reduction is therefore a challenging task, as it requires an understanding of a diverse range of predisposing factors, failure processes, and potential impacts over large spatial areas.

To date, the majority of studies focus on landslide hazard or susceptibility, constructed based on a statistical analysis of past landslide records (Calcaterra et al., 2003), on datasets describing typical landslide predisposing factors (van Westen et al., 2006; Reichenbach et al., 2018), or a combination of these two methods. Other studies have moved beyond this to evaluate landslide risk, combining hazard, vulnerability, and exposure (Lateltin et al., 2005; Cruden, 2018; Emberson et al., 2020). Vulnerability is the capacity to prevent, mitigate, or recover following a landslide (van Westen et al., 2006; Alexander, 1986; Chiocchio et al., 1997; Iovine and Parise, 2002; Chen et al., 2020), and it is commonly set at a constant value calibrated based on historical damage data or human development indicators (Atkinson, 2012). Exposure represents the spatial distribution of at-risk people, buildings, or other infrastructure.

It is instructive to compare the landslide risk assessment workflows to those from other hazards. For example, the exposure and vulnerability components of seismic risk maps are commonly drawn from building datasets (e.g., Coburn et al., 1992). This approach is justified as most earthquake deaths are directly associated with building collapse, and is appealing as it can be upscaled to large areas (Coburn et al., 1992; Doocy et al., 2013). Open-source building databases, such as OpenStreetMap, have recently improved the availability and accessibility of this information. Various studies have considered whether seismic hazard assessments may be directly applied to landslides, or whether an analogous approach may enable these building maps to be used as an input for landslide risk models (e.g., Pollock and Wartman, 2020; Jakob et al., 2012). In the case of catastrophe risk models applied by private sector insurance companies, exposure is commonly composed of an infrastructure map with a specific economic value associated with each property (Sterlacchini et al., 2007; Atkinson, 2012). However, for building databases to be directly applicable as exposure layers for landslides, we must evaluate whether they adequately capture not only the damage to buildings but also loss of life.

In this study, we investigate the relation between the reported total human loss from landslides (deaths and missing people) and reported building damage. We investigate this relation both on a global scale and through a detailed case study in Nepal. We consider whether building damage is a reliable proxy - with predictive value - for the total human loss from landslides, and what the implications of a decoupling between these two key indicators of impacts may be.

## 2 Methods

We use a harmonized database of disasters for 89 countries, DesInventar (Disaster Inventory System), to investigate the relation between landslide building damage and deaths (Atkinson, 2012; Yamazaki-Honda et al., 2019; Mazhin et al., 2021). DesInventar includes data from the mid-20th century to the present, with degree of completeness and metadata varying widely between countries. Nevertheless, it remains one of the most spatially and temporally comprehensive global-scale disaster databases (Yamazaki-Honda et al., 2019; Mazhin et al., 2021). Commonly used indicators for damage are recorded, including the number of buildings destroyed and damaged, the number of lives lost, and the number declared missing for each disaster (Yamazaki-Honda et al., 2019). There is substantial work on the differing degrees of damage that landslides can do to buildings (Alexander, 1986; Chiocchio et al., 1997; Iovine and Parise, 2002; Chen et al., 2020; Del Soldato et al., 2019), but these datasets do not allow for this level of detailed analysis and we simplify here to a binary damaged/not damaged classification.

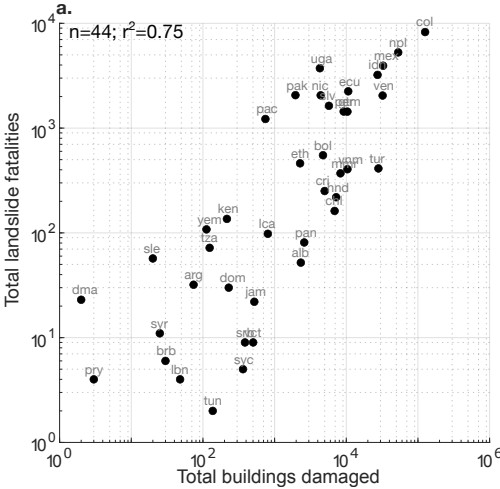 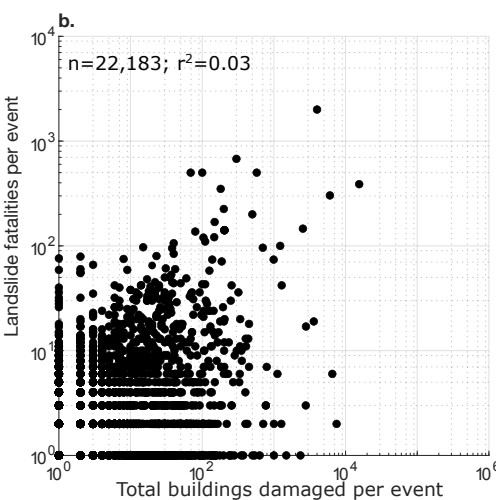

**Figure 1.** Plot of total human loss against total building damage for nationwide totals (a) and individual landslides (b). Country codes for in (a) are the standard nomenclatures from the DesInventar database and are included in the Supplemental Information.

For a nationwide case study of Nepal, we supplement the DesInventar event catalogue (which ends in 2013) with comparable data from the Nepal National Disaster Risk Reduction And Management Authority (from 2014) available through the Building Information Platform Against Disaster or BIPAD Portal (NDRRMA, 2020).

We begin with a systematic search of all keywords in the DesInventar database to identify words for 'landslide' in different languages (e.g., 'deslizamiento de tierra'), spelling errors (e.g., 'landside'), or different terms for a similar physical process (e.g., 'rock slide'). A full list of the keywords used is available in the Supplemental Information. We find a total of 47,213 individual landslides, of which we exclude 25,030 which resulted in neither deaths, missing people, or building damage and include 22,183 for further analysis which record losses in at least one of those categories (deaths, missing people, and/or

building damage). We regress the total human loss from each landslide (sum of deaths and number of missing persons) against the total building damage (sum of buildings destroyed and damaged) for each country with sufficient data (>10 landslides over the entire record), yielding estimates for a total of 44 countries. For Nepal, we run this workflow on both the DesInventar data alone and on a merged database comprising both DesInventar and BIPAD data.

We use different metrics to evaluate the power of total building damage (predictive variable) towards total human loss

(dependent variable): the coefficient of determination ($R^2$), the coefficient of estimation (CE), reduction of error (RE), and root mean squared error of prediction (RMSEP; Cook et al., 1994). The CE, RE, RMSEP, and relative RMSEP specifically test whether or not building damage can be used as a meaningful predictor for human damage. We adopt the formulae of Cook et al. (1994) to calculate the CE and RE, using bootstrapping to separate the data into 'validation' and 'calibration' datasets, randomly sampled (with repetition, both to the same size as the original dataset) 100 times to calculate the mean and standard

deviation for the CE and RE metrics. We calculate the RMSEP and relative RMSEP from the same bootstrapped datasets. To

contextualise the landslide results, we repeat the above analysis for several other disasters in the DesInventar database: floods, volcanic eruptions, earthquakes, and storms.

## 3 Results

### 3.1 Global analysis

On a global scale, total human loss positively correlates with total building damage from landslides. The linear fit is good when considering the nationwide averages ($R^2$=0.75, F-score=112, n=44, Figure 1a), but is negligible when disaggregating to individual events ($R^2$=0.03, F-score=1250, n=22,183 Figure 1b). The F-score confirms that the relationship between the two variables is statistically significant, despite the poor correlation. This poor goodness of fit remains when considering individual landslides for each of the 44 countries with >10 events, with a median $R^2$ of 0.0247 (interquartile range 0.0041, 0.1017) and relative RMSEP of 590% (interquartile range 276%, 1195%). For individual nations, the coefficient of estimation (median -0.22, interquartile range -4.14, -0.049) and reduction of error (median -0.049, interquartile range -0.18, 0.00) are also negative.

### 3.2 Case study: Nepal

In Nepal, an equally poor fit is apparent for the combined DesInventar and BIPAD disaster inventories (Figure 2a). The regression of total human loss against total building damage in this dataset of 3,206 landslides in Nepal has an $R^2$ of 0.0023 and an relative RMSEP of 1279 $\pm$ 197%. The coefficient of estimation (0.0023 $\pm$ 0.0031) and reduction of error (0.0028 $\pm$ 0.0032) are statistically indistinguishable from zero, indicating no predictive value. In simple terms, events causing the highest human loss are not significantly associated with those which cause the most building damage, and events destroying the most buildings are not always the deadliest.

### 3.3 Comparison with other disasters

We repeat the same analysis with different disasters from the DesInventar database. Some, such as floods and avalanches, also have a weak relationship between total human loss and total building damage. Conversely, other disasters such as tsunamis (n = 2,126; Figure 3b) and earthquakes (n = 19,180; Figure 3c) exhibit a strong link between high total building damage and high total human loss, with the greatest human loss concentrated in the events that also affected the most properties. For lightning strikes, we find the inverse result, with the highest total human loss in events affecting the least properties (n = 6,604, Figure 3d).

## 4 Discussions

The correlation between human loss and number of affected buildings from landlsides is good in national averages, but negligible when disaggregated to individual events. Both human loss and total building damage follow long-tailed distributions,

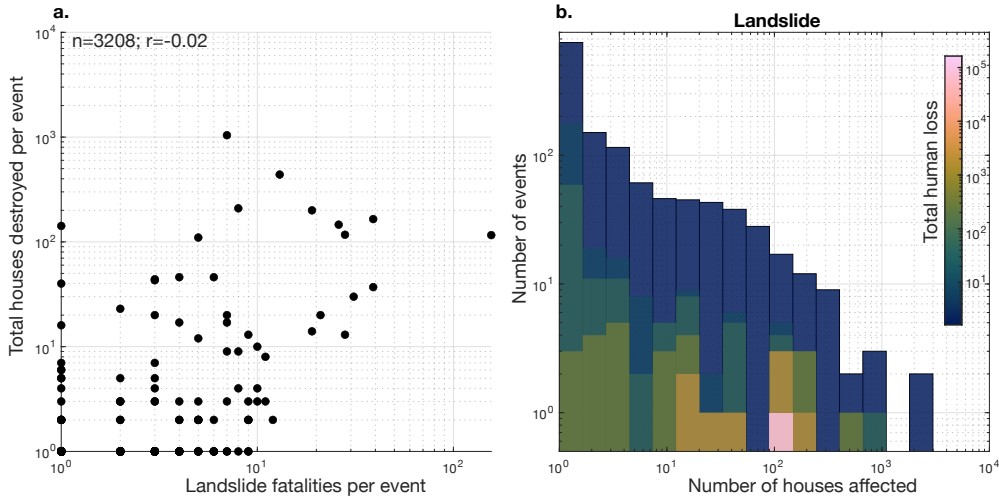

**Figure 2.** (a) Plot of human loss against total building damage for individual landslides in Nepal. (b) Histogram of total building damage in individual landslides for Nepal, with colours corresponding to associated total human loss. A strong correlation between building damage and human loss would result in a progressive left-to-right colour gradient and limited variation within each column. Instead, note that human loss within each column is highly variable.

with most damage being accounted for by a small number of events. However, in the case of landslides, we find a poor corre-
spondence between the events causing high levels of human loss and high levels of building damage. A comparison with other
disasters shows higher correlation between these extremes for other disaster types, such as earthquakes and tsunamis. Averag-
ing the impacts of multiple events, either as temporal averages (total damage per year) or geographic averages (total damage
per country) can mask this lack of correlation for individual events by cumulating both high-death, low-building damage events
and low-death high-building damage events.

All databases provide an imperfect and biased record of the impacts of disasters. Disasters will only be recorded if they are
large or damaging enough to be noteworthy, and the size of inventories varies drastically between the 89 countries included
in the DesInventar database (Yamazaki-Honda et al., 2019). Additionally, the number of deaths, missing people, and buildings
damaged may be either overestimated or underestimated for events that are recorded, and the accuracy of these estimates will
vary spatially and temporally (Yamazaki-Honda et al., 2019; Mazhin et al., 2021). Our study design partly mitigates these
limitations, as we investigate per-disaster damage instead of the total number of events, or total damage. Even though the data
sets are incomplete, we can still make inferences about the relation between total human loss and total building damage from
the events that were recorded. A comparison between landslides and other disasters provides a further test, as the relation
between building damage and human loss varies for different disasters. This relation is expected to be strong for earthquakes,
where a large proportion of human loss is directly caused by building collapse (Coburn et al., 1992; Doocy et al., 2013), and
particularly weak for lightning where buildings may shield inhabitants from damage. Both of these expectations are supported
by the DesInventar database (Supplemental Figure S1).

Our finding that damage to properties and human loss are poorly related is consistent with the complex and geographically dispersed nature of landslides and our current understanding of their links to human mortality. Different impacts may indeed be expected depending on landslide type: for instance, large landslides with clear warning signs may damage many buildings without loss of life, while small rockfalls may cause many fatalities without damaging any buildings. Similarly, landslide early warning systems may enable effective evacuations in some parts of the world, preventing fatalities but not building damage. The exact causes of this discrepancy, including different landslide processes, effective mitigation strategies, and spatially concentrated exposure and vulnerability, are likely to vary widely across the world and within this dataset. Pollock and Wartman (2020) showed the importance of demographic, situational, and particularly behavioural factors in determining landslide morbidity, arguing that the relationship between building damage and morbidity is therefore complex. A low correlation has previously been noted between landslide-related deaths and the economic cost of landslides (Hilker et al., 2009; Kennedy et al., 2015), although this study was limited to Switzerland alone. Creating an accurate risk map relies on a combination of two components: a map of the spatial distribution of the hazard, and a measure of the exposure to this hazard. This exposure layer will depend on the objective of the risk map; for instance, insurance disaster risk maps will often be based on infrastructure value maps. In the case of landslides, our results show that building maps or databases – for example, MSBuildings or OpenStreetMap – are an inadequate proxy for the total human loss from landslides, and should not be relied upon solely to estimate risk to human lives from landsliding.

Our results show that landslide disaster mitigation strategies using risk maps constructed from infrastructure assets or building datasets may implicitly prioritise monitoring or mitigation of high-building damage events instead of high total human loss events. This raises ethical and practical issues and is generally at odds with the primary objective of disaster risk reduction programs. Two different and complementary approaches stand out to improve our understanding and representation of human loss from landsliding. The first involves building a detailed understanding of local conditions through consultations and interviews, and the second involves large-scale (and ideally dynamic) population or exposure modelling. Both methods fall outside the traditional remit of landslide science, and highlight the need for transdisciplinary collaboration for effective landslide risk reduction.

Improving our understanding of disaster risk involves examining local risk perceptions, daily routine variations, and mitigation strategies. Local risk perceptions and actions may explain the low correlation between human casualties and building damage, either due to effective evacuation strategies reducing exposure or inadvertent actions increasing exposure (Pollock and Wartman, 2020). Additionally, considering local perspectives may reveal effective risk-reduction strategies and increase community involvement in mitigation efforts. The factors contributing to the low correlation between human casualties and the number of affected buildings are likely to vary across the 89 represented countries in the DesInventar database. Conducting informal and semi-structured interviews with local residents could help shed light on why building damage might exceed fatalities in certain landslides and vice versa within specific regions.

On a larger scale, population density and dynamic exposure maps offer an alternative perspective, albeit with some limitations. These population density maps often have low spatial resolution or rely on building data for interpolation (Lloyd et al., 2017). Additionally, static population density maps fail to capture substantial transient changes in population density occurring

on a daily, seasonal, or interannual basis. In scenarios such as landslide risk assessment, where evacuation may be impractical and vulnerability is high (Kennedy et al., 2015), dynamic exposure becomes a critical element of risk-to-life modelling. Innovative modelling approaches, such as agent-based models (e.g. Zayn et al., 2020), have the potential to account for spatiotemporal population movements across different timescales. However, these methods are relatively untested in the context of disaster risk reduction, presenting an open science challenge in need of further development.

## 5 Conclusions

This study shows a complex relationship between building damage and human loss from landslides. We find that, despite moderate correlation for national averages, the two are uncorrelated in individual events at both a global scale and in Nepal specifically. Therefore, building damage from landslides is not an effective predictor of the number of fatalities from the same event. Comparative analysis with other disasters highlights the contrasts between them, with a stronger link between building damage and human loss present for earthquakes and tsunamis, but not for other geohazards such as floods, avalanches, or lightning strikes. There is a need to develop exposure layers beyond simple building databases, encompassing localized insights into risk factors and dynamic population models, to improve mitigation of the deadliest landslides.

*Author contributions.* MV conceived the study and conducted the analyses with input from AD, AJ, EH, DL, and KA. All authors helped interpret the results and commented on the manuscript.

*Competing interests.* The authors declare no competing interests.

*Acknowledgements.* This research was supported by a grant from the UKRI Global Challenges Research Fund (NE/T01038X/1). We thank Daniel Costantini and one anonymous reviewer for their constructive comments, along with editor Mario Parise.

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
