# Peer review of "Brief Communication: Weak correlation between building damage and loss of life from landslides"

_Natural Hazards and Earth System Sciences, 2024_

## Author Comment (AC2)

**Brief Communication: Weak correlation between building damage and loss of life from landslides**

Maximillian Van Wyk de Vries[1,2,3], Alexandre Dunant[4], Amy L. Johnson[5], Erin L. Harvey[4], Sihan Li[6], Katherine Arrell[7], Jeevan Baniya[8], Dipak Basnet[8], Gopi K. Basyal[9], Nyima Dorjee Bhotia[8], Simon J. Dadson[3], Alexander L. Densmore[4], Tek Bahadur Dong[8], Mark E. Kincey[10], Katie Oven[7], Anuradha Puri[8], and Nick J. Rosser[4]

[1]Department of Geography, University of Cambridge, Cambridge CB2 3EL, UK.
[2]Department of Earth Sciences, University of Cambridge, Cambridge CB3 0EZ, UK.
[3]School of Geography and the Environment, University of Oxford, Oxford OX1 3QY, UK.
[4]Department of Geography, Durham University, Lower Mountjoy, South Rd, Durham DH13LE, UK.
[5]Department of Government and Sociology, Georgia College & State University, Milledgeville 31061, US.
[6]Department of Geography, University of Sheffield, Winter St, Sheffield S37ND, UK.
[7]Department of Geography and Environmental Sciences, Northumbria University, City Campus, Newcastle upon Tyne, NE1 7RU, UK.
[8]Social Science Baha, 345 Ramchandra Marg, Battisputali, Kathmandu, Nepal.
[9]National Society for Earthquake Technology, Sainbu Bhainsepati Residential Area, Lalitpur 13775, Nepal.
[10]School of Geography, Politics and Sociology, Newcastle University, Newcastle upon Tyne NE17RX, UK.

**Correspondence:** M. Van Wyk de Vries (msv27@cam.ac.uk)

**Response to Reviews:**

We thank both reviewers for their positive and constructive comments and for taking the time to evaluate our manuscript. We have responded to all points between the lines below in red.

Reviewer 1:

Interesting statistical analyses and results!

Below are a few comments that I have noticed:

- Lines 20-24: I would not say that seismic risk maps are constructed only on the basis of the type and condition of buildings. Rather, the seismic microzonation or the type and condition of the soil plays a central role (cf. various earthquakes such as the one in Emiglia Romagna or Abruzzo). I would suggest supplementing this.

Thank you for your comment. We agree that seismic risk assessments also consider seismic microzonation, including soil conditions and local geology, which significantly influence ground shaking and building response during earthquakes. However, our main point here is not to provide a full overview of possible contributions to seismic risk maps, but rather to highlight that the exposure and vulnerability elements are commonly drawn from building location and type databases. The seismic microzonation or the type and condition of the soil you mention will typically feature in the 'hazard' portion of the seismic risk. We will revise the manuscript to clarify this point. Specifically, we modify lines 20-24 to read:

"It is instructive to compare the assessment of risk from landslides to other hazards. For example, the exposure and vulnerability components of seismic risk maps are commonly drawn from datasets of building location and characteristics (e.g., Coburn et al., 1992). This approach is justified as most

earthquake deaths are directly associated with building collapse, and is appealing as it can be upscaled to large areas (Coburn et al., 1992; Doocy et al., 2013)."

- Lines 78-80: That's an interesting observation. However, I believe that this observation also has a lot to do with the type of process. For example, when a large slide is activated, you have often little human loss and a lot of damage to buildings. This is not the case with rockfalls, however, where there is often a high level of human loss and less damage to buildings, because these tend to be more localised. I would suggest that the type of landslide process must also be taken into account in any case.
- Further observation: It would be interesting to carry out such statistical analyses on databases that have a high level of detail in terms of geographical localisation and content/description of the event in order to compare whether the results are similar. One example is the IdroGEO platform in Italy - see, for example, the landslides of the Autonomous Province of Bolzano. We are at your complete disposal for a discussion of the analysis of our detailed datasets.

We appreciate your valuable comments and respond to both questions together. Indeed, we agree that one factor contributing to this discrepancy between human loss and building loss is the wide range of possible landslide processes. We have added a mention of this to the discussions but consider that the key message of this paper is that – regardless of landslide type, size, or location – the correlation between housing damage and deaths is poor. We hope that, by emphasizing this point, we can help deter future studies from inadvertently using building datasets to construct (human) landslide risk maps.

While we do not use a highly detailed dataset such as IdroGEO (another alternative being the UK's National Landslide Database) or bring in information about landslide type here, we agree that this would be a valuable addition to the study. It may, for instance, highlight some process-specific interactions (e.g. we might hypothesize about buildings offering more protection to certain types of landslides than others). Analyzing high-resolution datasets could indeed provide deeper insights and validate our findings at a more localized scale. While our current study focuses on global and national datasets due to availability, we recognize the value of detailed regional data.

An initial challenge is that the data used here, DesInventar and BiPAD, do not provide any further information about landslide type. While we do not think there is space to add this within the scope of this specific study and paper, we are interested in pursuing this as a follow-on project using these alternative databases and would gladly accept your offer of a discussion around this.

Reviewer 2

This study focuses on the relationship between building damage and human loss associated with landslides. By examining a global database, the authors show there is only a weak relationship between building damage and human loss for individual events. There is a much stronger correlation when data are averaged for individual countries. The lack of a strong correlation could have implications for the generation of exposure maps since these could be defined based on building databases. The paper is well written and easy to follow. I have mostly minor comments on methodology and potential discussion points.

We thank the reviewer for their positive assessment of our manuscript and respond to each of the points below.

1. Comparing and contrasting landslide and earthquake hazards in this context is interesting. One potential discussion point that is not mentioned is that there are early warning systems in place for some types of landslides in many countries but there are no analogous earthquake early warning systems. Since there are early warning systems for some landslides, could this contribute to a weaker correlation between building damage and human loss due to evacuation and improved awareness of the timing of the hazard? For example, we might expect there to be substantial building damage and minimal human loss in a residential area where there is a successful evacuation before a landslide. An exposure map made prior to this event that was based on a building database could still be a good indicator of loss of life in this case if the map was designed to represent a scenario without an evacuation. The potential for evacuations and increased awareness of the timing of the hazard could be a potential explanation for some of the points in figure 1b where there is substantial building damage and negligible human loss.

Thank you for highlighting this important consideration. You are correct that early warning systems and timely evacuations can significantly reduce human casualties in landslide events, potentially weakening the correlation between building damage and human loss. Unfortunately, none of the databases used here provide any indication of whether early warning systems were available for the events recorded.

We did conduct, in Nepal, a series of in-depth discussions with communities about landslides, although we did not consider the details of this possible to fit within this manuscript (requring too much background information for this short-format article). In Nepal, there are very few 'formal' landslide early warning systems, but there is equally much community level warning and self-adaptation that would lead to the same outcome. For instance, many people aware of the high landslide risk around their house would self-evacuate to family members' houses during times of heavy rainfall. We expect some of these factors to exist in different ways around much of the world – perhaps in other high-resource countries the role of 'technological' early warning systems is more substantial than that of self-evacuations and other mitigations (I think, for instance, of the recent evacuation at Brienz in Switzerland). We have slightly expanded our discussion of this in the manuscript, while also explaining that due to the complexity of the expected relationship (i.e. early warning systems not being the only, or necessarily even the most important factos) that it is out of the scope of this manuscript to fully explore it here. We mention this as a promising case to explore for future research.

2. There appear to be a number of points in figure 1b where there is human loss but negligible (i.e. 0 or 1) building damage. One interpretation is that these cases are associated with events in areas that are not residential, such as tourist locations or camping sites where there are people but no or very few permanent structures. I'm sure there could be a variety of other explanations for this as well, but discussing those possibilities could help place the weak correlation in better context since using a building database for exposure maps would only make sense and be possible in cases where there are buildings. When examining the strength of the correlation between building damage and human loss within the context of implications for exposure maps, it would be ideal to only include events that occurred in areas with

permanent structures. I understand that this is probably not feasible with the type of data being used here but some discussion around this and related points could be helpful.

You raise an excellent point regarding events with fatalities but minimal building damage. These instances may occur in areas without permanent structures, such as roads, hiking trails, or temporary settlements like campsites, where people are exposed to landslides. We agree that it is evident that building databases may not adequately represent exposure in these contexts. There are a number of examples of this from the recent monsoon in Nepal where fatal landslides occurred on roads and other non-built up areas (e.g. https://www.bbc.co.uk/news/articles/c978l5mpqe7o). If a building dataset were used to construct an exposure layer, the exposure would be zero in these areas (this is, for instance, common for seismic risk analyses). Therefore we consider that it is important to retain all events and not only those which occurred in built-up areas, or else we will potentially remove a category of high-fatality landslides events that are important to consider. Furthermore, as you mention, distinguishing between the two types would not be possible using only the disaster datasets available here.

Line 51: Is this a linear regression as mentioned in the results? Why only explore a linear relationship between building damage and human loss? Based on figure 2a, a power law might fit better and could be worth exploring. If not, it could also be interesting to report a spearman correlation coefficient. Either way, could you elaborate on the reasoning for expecting a linear relationship if that is the case.

Thank you for this thoughtful comment. We initially used linear regression for simplicity and to provide a straightforward interpretation of the relationship between building damage and human loss. Indeed, a linear relationship would be expected if (i) there is a relatively constant distribution of people in each building and (ii) damage to buildings is commonly associated with deaths. We calculate a Spearman correlation of 0.91 for the countrywise aggregated data and 0.044 for the full dataset, but are unsure that the rank-correlation is more informative here (i.e. it has more potential to mask extreme discrepancies in some event types).

Line 66: Can you comment on the performance of the linear fit based on the distribution of the residuals?

We have evaluated the residuals of the linear fit – a histogram of these is attached below. Here, we have conducted 1000 bootstrapped linear fit attempts and computed a separate residual for each in order to account for uncertainty. We note that there are also extreme outliers which are not visualised in this plot – the scatterplot shown below provides some idea of these. The residuals are close to normally distributed with a slight skew towards positive values in some iterations (likely associated with the differing effect of outliers). Overall, the residual analysis does not provide strong evidence for considering alternative, non-linear models.

[Figure]

Line 87-90: This gets back to the above comments about potential explanations for high-death, low building damage events (e.g. events that occur in areas without permanent structures where building databases wouldn't be a reasonable possibility for creating an exposure map) and low-death, high building damage events (e.g. events where an evacuation or early warning minimized human loss). I think this is worth discussing since these types of events seem to have a strong effect on the strength of the correlation.

We fully agree about the importance of high-death/low-damage events and vice versa on the overall results. As mentioned above, we have explored some of the reasons for this in more detail cannot fit this extended discussion within this paper. We have, however, expanded our discussion of how these might affect the fit to the data. Indeed, it is specifically this subset of high-death/low-damage disasters that we are most concerned could be missed if the discrepancy between building damage and fatalities is not fully considered.

Line 141-143: The strength of the correlation for other hazards is mentioned in the conclusions but not in the results section. I think it would be appropriate to add a little more about these other correlations to the results section.

Thank you for this suggestion. We will revise the results section to include a more detailed discussion of the correlation between building damage and human loss for other hazards. Including this information will strengthen our argument by providing quantitative evidence of how landslides differ from other hazards in this context.

Once again, we thank both reviewers for their time and comments which have substantially improved our manuscript.